# FLATTEN: OPTICAL FLOW-GUIDED ATTENTION FOR CONSISTENT TEXT-TO-VIDEO EDITING

**Yuren Cong**[1,2*]**, Mengmeng Xu**[2]**, Christian Simon**[2]**, Shoufa Chen**[3]**, Jiawei Ren**[4]**,
Yanping Xie**[2]**, Juan-Manuel Perez-Rua**[2]**, Bodo Rosenhahn**[1]**, Tao Xiang**[2]**, Sen He**[2]

[1]Leibniz University Hannover, [2]Meta AI, [3]The University of Hong Kong, [4]Nanyang Technological University

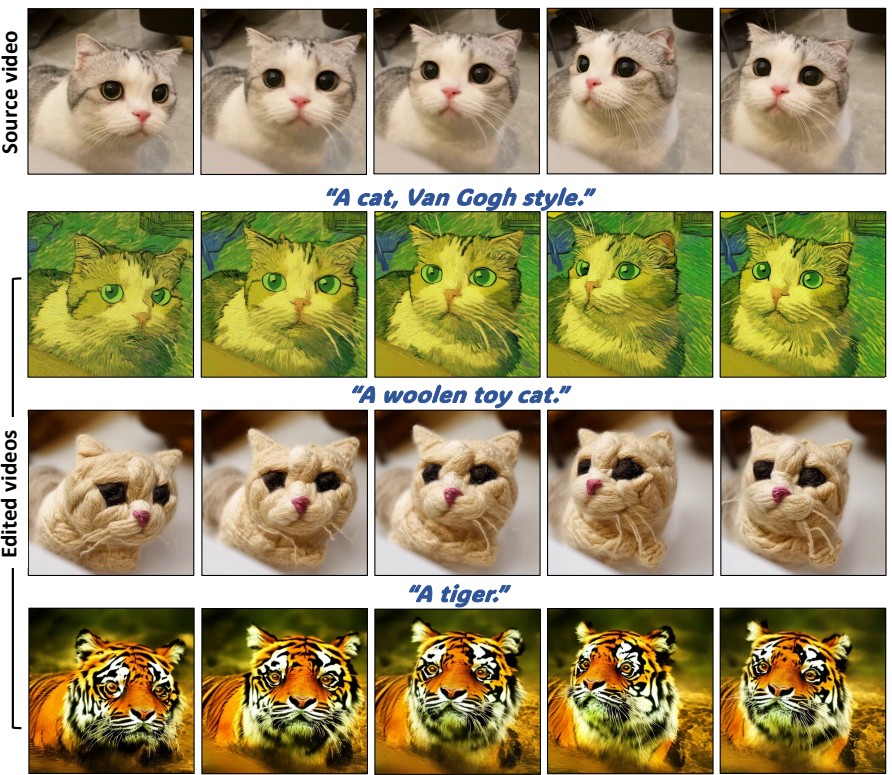

**Figure 1:** Our method generates visually consistent videos that adhere to different types (style, texture, and category) of textual prompts while faithfully preserving the motion in the source video.

## ABSTRACT

Text-to-video editing aims to edit the visual appearance of a source video conditional on textual prompts. A major challenge in this task is to ensure that all frames in the edited video are visually consistent. Most recent works apply advanced text-to-image diffusion models to this task by inflating 2D spatial attention in the U-Net into spatio-temporal attention. Although temporal context can be added through spatio-temporal attention, it may introduce some irrelevant information for each patch and therefore cause inconsistency in the edited video. In this paper, for the first time, we introduce optical flow into the attention module in the diffusion model's U-Net to address the inconsistency issue for text-to-video editing. Our method, **FLATTEN**, enforces the patches on the same flow path across different frames to attend to each other in the attention module, thus improving the visual consistency in the edited videos. Additionally, our method is training-free and can be seamlessly integrated into any diffusion-based text-to-video editing methods and improve their visual consistency. Experiment results on existing text-to-video editing benchmarks show that our proposed method achieves the new state-of-the-art performance. In particular, our method excels in maintaining the visual consistency in the edited videos. The project page is available at https://flatten-video-editing.github.io/.

---

*Work done during an internship at Meta AI.

# 1 INTRODUCTION

Short videos have become increasingly popular on social platforms in recent years. To attract more attention from subscribers, people like to edit their videos to be more intriguing before uploading them onto their personal social platforms. Text-to-video (T2V) editing, which aims to change the visual appearance of a video according to a given textual prompt, can provide a new experience for video editing and has the potential to significantly increase flexibility, productivity, and efficiency. It has, therefore, attracted a great deal of attention recently (Wu et al., 2022; Khachatryan et al., 2023; Qi et al., 2023; Zhang et al., 2023; Ceylan et al., 2023; Qiu et al., 2023; Ma et al., 2023).

A critical challenge in text-to-video editing compared to text-to-image (T2I) editing is visual consistency, *i.e.*, the content in the edited video should have a smooth and unchanging visual appearance throughout the video. Furthermore, the edited video should preserve the motion from the source video with minimal structural distortion. These challenges are expected to be alleviated by using fundamental models for text-to-video generation (Ho et al., 2022a; Singer et al., 2022; Blattmann et al., 2023; Yu et al., 2023). Unfortunately, these models usually take substantial computational resources and gigantic amounts of video data, and many models are unavailable to the public.

Most recent works (Wu et al., 2022; Khachatryan et al., 2023; Qi et al., 2023; Zhang et al., 2023; Ceylan et al., 2023) attempt to extend the existing advanced diffusion models for text-to-image generation to a text-to-video editing model by inflating spatial self-attention into spatio-temporal self-attention. Specifically, the features of the patches from different frames in the video are combined in the extended spatio-temporal attention module, as depicted in Figure 2. By capturing spatial and temporal context in this way, these methods require only a few fine-tuning steps or even no training to accomplish T2V editing. Nevertheless, this simple inflation operation introduces irrelevant information since each patch attends to all other patches in the video and aggregates their features in the dense spatio-temporal attention.

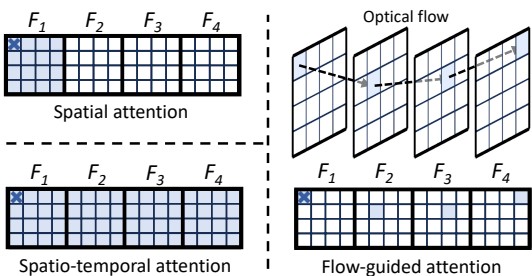

**Figure 2:** Illustration of spatial attention, spatio-temporal attention, and our flow-guided attention. The patches marked with the crosses attend to the colored patches and aggregate their features. $F_k$ indicates the feature map of the $k$-th video frame.

The irrelevant patches in the video can mislead the attention process, posing a threat to the consistency control of the edited videos. As a result, these approaches still fall short of the visual consistency challenge in text-to-video editing.

In this paper, for the first time, we propose **FLATTEN**, a novel (optical) FLow-guided ATTENtion that seamlessly integrates with text-to-image diffusion models and implicitly leverages optical flow for text-to-video editing to address the visual consistency limitation in previous works. FLATTEN enforces the patches on the same flow path across different frames to attend to each other in the attention module, thus improving the visual consistency of the edited video. The main advantage of our method is that enables the information to communicate accurately across multiple frames guided by optical flow, which stabilizes the prompt-generated visual content of the edited videos. More specifically, we first use a pre-trained optical flow prediction model (Teed & Deng, 2020) to estimate the optical flow of the source video. The estimated optical flow is then used to compute the trajectories of the patches and guide the attention mechanism between patches on the same trajectory. Meanwhile, we also propose an effective way to integrate flow-guided attention into the existing diffusion process, which can preserve the per-frame feature distribution, even without any training. We present a T2V editing framework utilizing FLATTEN as a foundation and employing T2I editing techniques such as DDIM inversion (Mokady et al., 2023) and feature injection (Tumanyan et al., 2023). We observe high-quality and highly consistent text-to-video editing, as shown in Figure 1. Furthermore, our proposed method can be easily integrated into other diffusion-based text-to-video editing methods and improve the visual consistency of their edited videos.

**The contributions** of this work are as follows: (1) We propose a novel flow-guided attention (FLATTEN) that enables the patches on the same flow path across different frames to attend to each other

during the diffusion process and present a framework based on FLATTEN for high-quality and highly consistent T2V editing. (2) Our proposed method, FLATTEN, can be easily integrated into existing text-to-video editing approaches without any training or fine-tuning to improve the visual consistency of their edited results. (3) We conduct extensive experiments to validate the effectiveness of our method. Our model achieves the new state-of-the-art performance on existing text-to-video editing benchmarks, especially in maintaining visual consistency.

## 2 RELATED WORK

**Image and Video Generation**   Image generation is a popular generative task in computer vision. Deep generative models, e.g., GAN (Karras et al., 2019; Kang et al., 2023) and auto-regressive Transformers (Ding et al., 2021; Esser et al., 2021; Yu et al., 2022) have demonstrated their capacity. Recently, diffusion models (Ho et al., 2020; Song et al., 2020a;b) have received much attention due to their stability. Many T2I generation methods based on diffusion models have emerged and achieved superior performance (Ramesh et al., 2021; 2022; Saharia et al., 2022; Balaji et al., 2022). Some of these methods operate in pixel space, while others work in the latent space of an auto-encoder.

Video generation (Le Moing et al., 2021; Ge et al., 2022; Chen et al., 2023a; Cong et al., 2023; Yu et al., 2023; Luo et al., 2023) can be viewed as an extension of image generation with additional dimension. Recent video generation models (Singer et al., 2022; Zhou et al., 2022; Ge et al., 2023) attempt to extend successful text-to-image generation models into the spatio-temporal domain. VDM (Ho et al., 2022b) adopt a spatio-temporal factorized U-Net for denoising while LDM (Blattmann et al., 2023) implement video diffusion models in the latent space. Recently, controllable video generation (Yin et al., 2023; Li et al., 2023; Chen et al., 2023b; Teng et al., 2023) guided by optical flow fields facilitates dynamic interactions between humans and generated content.

**Text-to-Image Editing**   T2I editing is the task of editing the visual appearance of a source image based on textual prompts. Many recent methods (Avrahami et al., 2022; Couairon et al., 2022; Zhang & Agrawala, 2023) work on pre-trained diffusion models. SDEdit (Meng et al., 2021) adds noise to the input image and performs denoising through the specific prior. Pix2pix-Zero (Parmar et al., 2023) performs cross-attention guidance while Prompt-to-Prompt (Hertz et al., 2022) manipulates the cross-attention layers directly. PNP-Diffusion (Tumanyan et al., 2023) saves diffusion features during reconstruction and injects these features during T2I editing. While video editing can benefit from these creative image methods, relying on them exclusively can lead to inconsistent output.

**Text-to-Video Editing**   Gen-1 (Esser et al., 2023) demonstrates a structure and content-driven video editing model while Text2Live (Bar-Tal et al., 2022) uses a layered video representation. However, training these models is very time-consuming. Recent works attempt to extend pre-trained image diffusion models into a T2V editing model. Tune-A-Video (Wu et al., 2022) extends a latent diffusion model to the spatio-temporal domain and fine-tunes it with source videos, but still has difficulties in modeling complex motion. Text2Video-Zero (Khachatryan et al., 2023) and ControlVideo (Zhang et al., 2023) use ControlNet (Zhang & Agrawala, 2023) to help editing. They can preserve the per-frame structure but relatively lack control of visual consistency. FateZero (Qi et al., 2023) introduces an attention blending block to enhance shape-aware editing while the editing words have to be specified. To improve consistency, TokenFlow (Geyer et al., 2023) enforces linear combinations between diffusion features based on source correspondences. However, the pre-defined combination weights are not adapted to all videos, resulting in high-frequency flickering.

Different from the aforementioned methods, we propose a novel flow-guided attention, which implicitly uses optical flow to guide attention modules during the diffusion process. Our framework can improve the overall visual consistency for T2V editing and can also be seamlessly integrated into existing video editing frameworks without any training or fine-tuning.

## 3 METHODOLOGY

### 3.1 PRELIMINARIES

**Latent Diffusion Models**   Latent diffusion models operate in the latent space with an auto-encoder and demonstrate superior performance in text-to-image generation. In the forward process, Gaussian

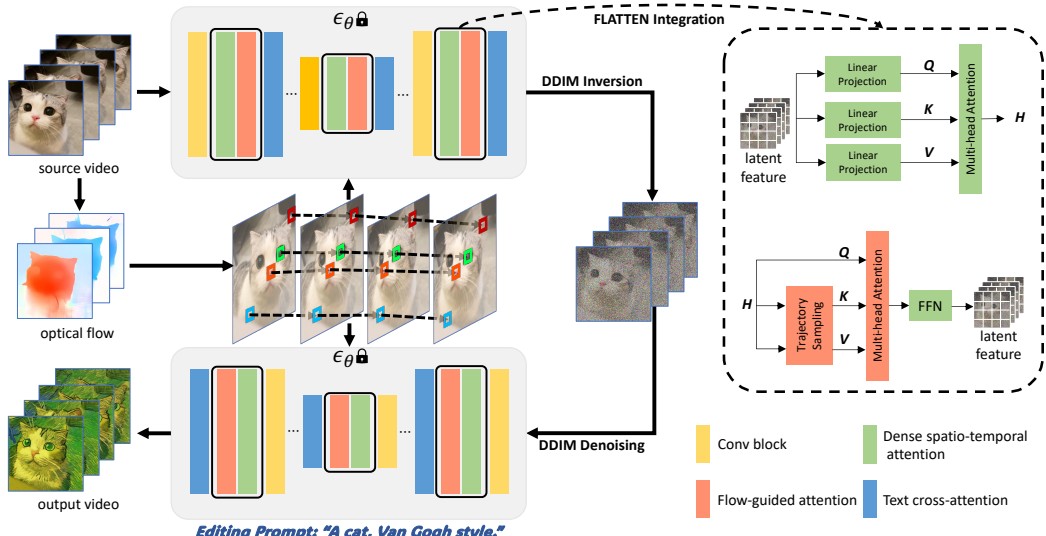

**Figure 3:** Overview of our framework. We inflate the existing U-Net architecture along the temporal axis and combine flow-guided attention (FLATTEN) with dense spatio-temporal attention to avoid introducing any new parameters. The outcome of dense spatio-temporal attention $\boldsymbol{H}$ is further used for FLATTEN. The keys and values for FLATTEN are gathered from $\boldsymbol{H}$ based on the patch trajectories sampled from the optical flow. The weights of the U-Net $\epsilon_\theta$ are frozen.

noise is added to the latent input $\boldsymbol{z}_0$. The density of $\boldsymbol{z}_t$ given $\boldsymbol{z}_{t-1}$ can be formulated as:

$$q(\boldsymbol{z}_t|\boldsymbol{z}_{t-1}) = \mathcal{N}(\boldsymbol{z}_t; \sqrt{1-\beta_t}\boldsymbol{z}_{t-1}, \beta_t\mathbf{I}), \tag{1}$$

where $\beta_t$ is the variance schedule for the timestep $t$. The number of timesteps used to train the diffusion model is denoted by $T$. The backward process uses a trained U-Net $\epsilon_\theta$ for denoising:

$$p_\theta(\boldsymbol{z}_{t-1}|\boldsymbol{z}_t) = \mathcal{N}(\boldsymbol{z}_{t-1}; \mu_\theta(\boldsymbol{z}_t, \boldsymbol{\tau}, t), \Sigma_\theta(\boldsymbol{z}_t, \boldsymbol{\tau}, t)), \tag{2}$$

where $\boldsymbol{\tau}$ indicates the textual prompt. $\mu_\theta$ and $\Sigma_\theta$ are computed by the denoising model $\epsilon_\theta$.

**DDIM Inversion** DDIM can convert a random noise to a deterministic $\boldsymbol{z}_0$ during sampling (Song et al., 2020a; Dhariwal & Nichol, 2021). Based on the assumption that the ODE process can be reversed in the small-step limit, the deterministic DDIM inversion can be formulated as:

$$\boldsymbol{z}_{t+1} = \sqrt{\frac{\alpha_{t+1}}{\alpha_t}}\boldsymbol{z}_t + \sqrt{\alpha_{t+1}}\left(\sqrt{\frac{1}{\alpha_{t+1}-1}} - \sqrt{\frac{1}{\alpha_t}-1}\right)\epsilon_\theta(\boldsymbol{z}_t), \tag{3}$$

where $\alpha_t$ denotes $\prod_{i=1}^{t}(1-\beta_i)$. DDIM inversion is employed to invert the input $\boldsymbol{z}_0$ into $\boldsymbol{z}_T$, which can be used for reconstruction and further editing tasks.

## 3.2 OVERALL FRAMEWORK

Our framework aims to edit the source video $\mathcal{V}$ according to an editing textual prompt $\boldsymbol{\tau}$ and output a visually consistent video. To this end, we expand the U-Net architecture of a T2I diffusion model along the temporal axis inspired by previous works (Wu et al., 2022; Khachatryan et al., 2023; Zhang et al., 2023). Furthermore, to facilitate consistent T2V editing, we incorporate flow-guided attention (FLATTEN) into the U-Net blocks without introducing new parameters. To retain the high-fidelity of the generated video, we employ DDIM inversion in the latent space with our re-designed U-Net to estimate the latent noise $\boldsymbol{z}_T$ from the source video. We use empty text for DDIM inversion without the need to define a caption for the source video. Lastly, we generate an edited video using the DDIM process with inputs from the latent noise $\boldsymbol{z}_T$ and the target prompt $\boldsymbol{\tau}$. Our framework as illustrated in Figure 3 is training-free, thus comfortably reducing additional computation.

**U-Net Inflation**   The original U-Net architecture employed in an image-based diffusion model comprises a stack of 2D convolutional residual blocks, spatial attention blocks, and cross-attention blocks that incorporate textual prompt embeddings. To adapt the T2I model to the T2V editing task, we inflate the convolutional residual blocks and the spatial attention blocks. Similar to previous works (Ho et al., 2022b; Wu et al., 2022), the $3 \times 3$ convolution kernels in the convolutional residual blocks are converted to $1 \times 3 \times 3$ kernels by adding a pseudo temporal channel. In addition, the spatial attention is replaced with a dense spatio-temporal attention paradigm. In contrast to the spatial self-attention strategy applied to the patches in a single frame, we adopt all patch embeddings across the entire video as the queries ($Q$), keys ($K$), and values ($V$). This dense spatio-temporal attention can provide a comprehensive perspective throughout the video. Note that the parameters of the linear projection layers and the feed-forward networks in the new dense spatio-temporal attention blocks are inherited from those in the original spatial attention blocks.

**FLATTEN Integration**   To further improve the visual consistency of the output frames, we integrate our proposed flow-guided attention in the extended U-Net blocks. We combine FLATTEN with dense spatio-temporal attention since both attention mechanisms are designed to aggregate visual context. Given the latent video features, we first perform dense spatio-temporal attention. Specific linear projection layers are employed to convert the patch embeddings of the latent features into the queries, keys, and values, respectively. The results of dense spatio-temporal attention are denoted as $H$. To avoid introducing newly trainable parameters and preserve the feature distribution, we do not apply new linear transformations to recompute the queries, keys, and values. We directly use $H$ as the input of flow-guided attention. Note that no positional encoding is introduced. When a patch embedding serves as a query, the corresponding keys and the values for FLATTEN are gathered from the output of dense spatio-temporal attention $H$ based on the patch trajectories sampled from optical flow. More details are demonstrated in Section 3.3. After performing flow-guided attention, the output is forwarded to the feed-forward network from the dense spatio-temporal attention block. We activate FLATTEN not only during DDIM sampling but also when performing DDIM inversion since using FLATTEN in DDIM inversion allows a more efficient inversion by introducing additional temporal dependencies. More details are discussed in Appendix A.

We also implement the feature injection following the image editing method (Tumanyan et al., 2023). For efficiency, we do not reconstruct the source video but inject the features from DDIM inversion during sampling. With these adaptations, our framework establishes and enhances the connections between frames, thus contributing to high-quality and highly consistent edited videos.

## 3.3   FLOW-GUIDED ATTENTION

**Optical Flow Estimation**   Given two consecutive RGB frames from the source video, we use RAFT (Teed & Deng, 2020) to estimate optical flow. The optical flow between two frames denotes a dense pixel displacement field $(f_x, f_y)$. The coordinates of each pixel $(x_k, y_k)$ in the $k$-th frame can be projected to its corresponding coordinates in the $(k+1)$-th frame based on the displacement field. The new coordinates in the $(k+1)$-th frame can be formulated as:

$$(x_{k+1}, \ y_{k+1}) = (x_k + f_x(x_k, y_k), \ y_k + f_y(x_k, y_k)). \tag{4}$$

In order to implicitly use optical flow to guide the attention modules, we downsample the displacement fields of all frame pairs to the resolution of the latent space.

**Patch Trajectory Sampling**   We sample the patch trajectories in the latent space based on the downsampled fields $(\hat{f}_x, \hat{f}_y)$. We start iterating from the patches on the first frame. For a patch with coordinates $(x_0, y_0)$ on the first frame, its coordinates on all subsequent frames can be derived from the displacement field. The coordinates are linked, and the trajectory sequence can be presented as:

$$traj = \{(x_0, y_0), (x_1, y_1), (x_2, y_2), \cdots, (x_K, y_K)\}, \tag{5}$$

where $K$ denotes the frame number of the source video. For a latent space with the size $H \times W$, there is ideally a trajectory set denoted as $\{traj_1, traj_2, ..., traj_N\}$, where $N = HW$. However, certain patches disappear over time, and new patches appear in the video. For each new patch that appears in the video, a new trajectory is created. As a result, the size of the trajectory set $N$ is generally larger than $HW$. To simplify the implementation of flow-guided attention, when an occlusion happens, we randomly select a trajectory to continue sampling and stop the other conflicting trajectories. This strategy ensures that each patch in the video is uniquely assigned to a single trajectory, and there is no case where a patch is on multiple trajectories.

**Figure 4:** Illustration of FLATTEN. We use RAFT to estimate the optical flow of the source video and downsample them to the resolution of the latent space. The trajectories of the patches in the latent space are sampled based on the displacement field. For each query, we gather the patch embeddings on the same trajectory from the latent feature as the corresponding key and value. The multi-head attention is then performed, and the patch embeddings are updated.

**Attention Process**    Flow-guided attention is performed on the sampled patch trajectories. The overview of FLATTEN is illustrated in Figure 4. We gather the embeddings of the patches on the same trajectory from the latent feature $z$. The patch embeddings on a trajectory $traj$ can be presented as:

$$z_{traj} = \{z(x_0, y_0), z(x_1, y_1), z(x_2, y_2), \cdots, z(x_K, y_K)\}, \tag{6}$$

where $z(x_k, y_k)$ indicates the patch embedding at the coordinates $(x_k, y_k)$ in the $k$-th frame. We perform multi-head attention with the patch embeddings on the same trajectory. For a query $z(x_k, y_k)$, the corresponding keys and values are the other patch embeddings on the same trajectory $traj$. No additional position encoding is introduced. Our flow-guided attention can be formulated as follows:

$$\boldsymbol{Q} = \boldsymbol{z}(x_k, y_k), \tag{7}$$

$$\boldsymbol{K} = \boldsymbol{V} = \boldsymbol{z}_{traj} - \{\boldsymbol{z}(x_k, y_k)\}, \tag{8}$$

$$\text{Attn}(\boldsymbol{Q}, \boldsymbol{K}, \boldsymbol{V}) = \text{Softmax}(\frac{\boldsymbol{Q}\boldsymbol{K}^T}{\sqrt{d}})\boldsymbol{V}, \tag{9}$$

where $\sqrt{d}$ is a scaling factor. The latent features $z$ are updated by flow-guided attention to eliminate the negative effects from feature aggregation of irrelevant patches in dense spatio-temporal attention. Importantly, we ensure that each patch embedding on the latent feature is uniquely assigned to a single trajectory during patch trajectory sampling. This assignment resolves conflicts and allows for a comprehensive update of all patch embeddings.

We utilize optical flow to connect the patches in different frames and sample the patch trajectories. Our flow-guided attention facilitates the information exchange between patches on the same trajectory, thus improving visual consistency in video editing. We integrate FLATTEN into our framework and implement text-to-video editing without any additional training. Furthermore, FLATTEN can also be easily integrated into any diffusion-based T2V editing method, as shown in Section 4.4.

## 4    EXPERIMENTS

### 4.1    EXPERIMENTAL SETTINGS

**Datasets**    We evaluate our text-to-video editing framework with 53 videos sourced from LOVEU-TGVE*. 16 of these videos are from DAVIS (Perazzi et al., 2016), and we denote this subset as TGVE-D. The other 37 videos are from Videvo, which are denoted as TGVE-V. The resolution of the videos is re-scaled to $512 \times 512$. Each video consists of 32 frames labeled with a ground-truth caption and 4 creative textual prompts for editing.

**Evaluation Metrics**    As per standard (Wu et al., 2022; Qi et al., 2023; Ceylan et al., 2023; Geyer et al., 2023), we use the following automatic evaluation metrics: For textual alignment, we use CLIP (Radford et al., 2021) to measure the average cosine similarity between the edited frames and the textual prompt, denoted as CLIP-T. To evaluate visual consistency, we adopt the flow warping error $E_{warp}$ (Lai et al., 2018), which warps the edited video frames according to the estimated optical

---

*https://sites.google.com/view/loveucvpr23/track4

**Table 1:** Quantitative results on TGVE-D and TGVE-V.

| Method | TGVE-D | | | | | TGVE-V | | | | |
|---|---|---|---|---|---|---|---|---|---|---|
| | CLIP-F $\uparrow$ | PickScore $\uparrow$ | CLIP-T $\uparrow$ | $E_{warp} \downarrow$ | $S_{edit} \uparrow$ | CLIP-F $\uparrow$ | PickScore $\uparrow$ | CLIP-T $\uparrow$ | $E_{warp} \downarrow$ | $S_{edit} \uparrow$ |
| Tune-A-Video | 91.05 | 20.58 | 27.33 | 29.23 | 9.35 | 96.30 | 20.20 | 25.84 | 15.38 | 16.80 |
| Text2Video-Zero | 92.39 | 20.32 | 27.86 | 22.07 | 12.62 | **96.84** | 20.43 | 26.53 | 11.55 | 22.97 |
| ControlVideo | 91.68 | 20.56 | 27.72 | 6.81 | 40.70 | 96.55 | 20.36 | 25.92 | 6.32 | 41.01 |
| FateZero | **92.58** | 20.45 | 27.06 | 5.79 | 46.74 | 96.64 | 20.09 | 25.72 | 5.10 | 50.43 |
| TokenFlow | 92.45 | 20.93 | 26.91 | 5.36 | 50.21 | 96.72 | 20.61 | 25.57 | **3.15** | 81.17 |
| FLATTEN (ours) | 92.49 | **20.95** | **28.05** | **4.92** | **57.01** | 96.75 | **20.63** | **26.70** | 3.16 | **84.49** |

flow of the source video and measures the pixel-level difference. Using these metrics independently cannot comprehensively represent editing performance. For instance, $E_{warp}$ reports 0 errors when the edited video is exactly the source video. Therefore, we propose $S_{edit}$ as our main evaluation metric, which combines CLIP-T and $E_{warp}$ as a unified score. Specifically, the editing score is calculated as $S_{edit}$ = CLIP-T/$E_{warp}$. Following the previous work (Wu et al., 2022), we also adopt CLIP-F and PickScore, which computes the average cosine similarity between all frames in a video and the estimated alignment with human preferences, respectively. For brevity, the numbers of CLIP-F/CLIP-T/$E_{warp}$ shown in this paper are scaled up by 100/100/1000.

**Implementation Details** We inflate a pre-trained text-to-image diffusion model and integrate FLATTEN into the U-Net to implement T2V editing without any training or fine-tuning. To estimate the optical flow of the source videos, we utilize RAFT (Teed & Deng, 2020). We find that applying flow-guided attention in DDIM inversion can also improve latent noise estimation by introducing additional temporal dependencies. Therefore, we use flow-guided attention both in DDIM sampling and inversion. More details are shown in Appendix A. We implement 100 timesteps for DDIM inversion and 50 timesteps for DDIM sampling. Following the image editing method (Tumanyan et al., 2023), the diffusion features are saved during DDIM inversion and are further injected during sampling. To efficiently perform the dense spatio-temporal attention in the modified U-Net, we use xFormers (Lefaudeux et al., 2022), which can reduce GPU memory consumption.

## 4.2 QUANTITATIVE COMPARISON

We compare our approach with 5 publicly available text-to-video editing methods: Tune-A-Video (Wu et al., 2022), FateZero (Qi et al., 2023), Text2Video-Zero (Khachatryan et al., 2023), ControlVideo (Zhang et al., 2023), and TokenFlow (Geyer et al., 2023). In these methods, Tune-A-Video requires fine-tuning the source videos. Both Tune-A-Video and FateZero need the additional caption of the source video, while our model does not. Text2Video-Zero and ControlVideo use ControlNet (Zhang & Agrawala, 2023) to preserve the structural information. Edge maps are used as the condition in our experiments, which have better performance than depth maps. TokenFlow linearly combines the diffusion features based on the correspondences of the source video features.

Table 1 shows the quantitative comparisons of TGVE-D and TGVE-V. Our approach outperforms other compared methods in terms of CLIP-T, PickScore, and editing score $S_{edit}$ on both datasets. In terms of the warping error $E_{warp}$, our method is slightly $0.1 \times 10^{-3}$ lower than TokenFlow. While considering textual faithfulness, our CLIP-T score is significantly higher. As a result, our method has a higher editing score overall. Text2Video-Zero has high CLIP-F and CLIP-T, but performs weakly in terms of visual consistency. Although FateZero has the highest CLIP-F on TGVE-D, its output video is sometimes very similar to the source video due to the hyperparameter setting issue. Our approach demonstrates superior performance on all evaluation metrics.

## 4.3 QUALITATIVE RESULTS

The qualitative comparison is presented in Figure 5. The source video at the top is from TGVE-D, and the source video at the bottom is from TGVE-V. Tune-A-Video generates videos with high quality per frame, but it struggles to preserve the source structure, *e.g.*, the wrong number of trucks. FateZero sometimes cannot edit the visual appearance based on the prompt, and the output video is almost identical to the source, as shown in the top example. Both Text2Video-Zero and ControlVideo rely on pre-existing features (*e.g.*, edge maps) provided by ControlNet. If the source condition features are of low quality, for example, due to motion blur, this leads to an overall decrease in video editing quality. TokenFlow samples keyframes and performs a linear combination of features to keep visual consistency. However, the pre-defined combination weights may not be appropriate for all videos. In the example at the bottom, a white sun intermittently appears and disappears in the

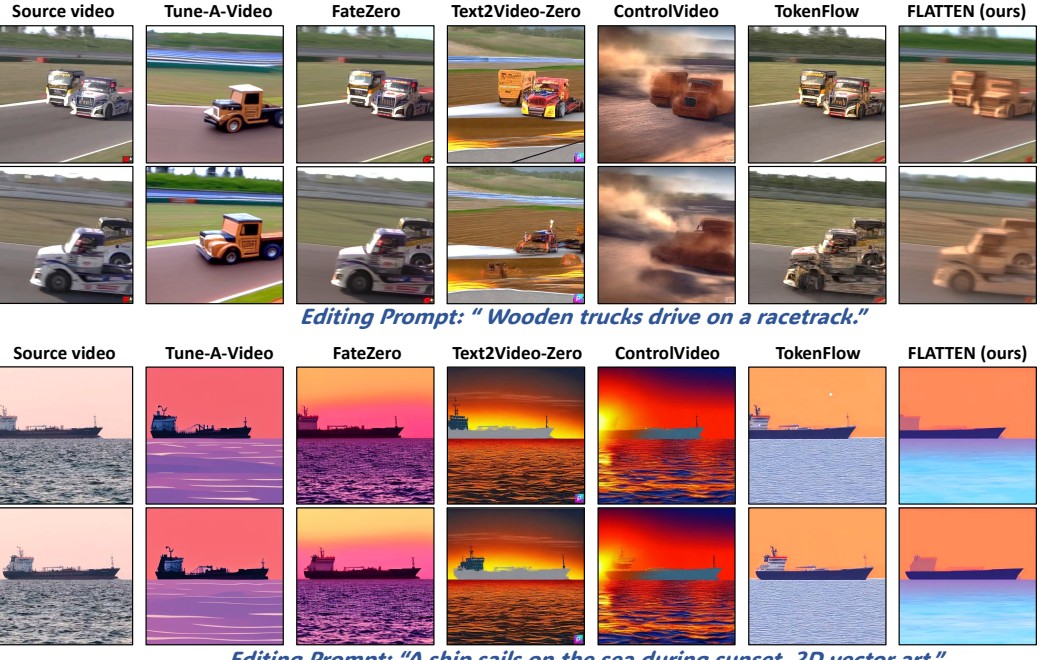

*Editing Prompt: " Wooden trucks drive on a racetrack."*

*Editing Prompt: "A ship sails on the sea during sunset, 2D vector art."*

**Figure 5:** Qualitative comparison between advanced T2V editing approaches and our method. The first column shows the source frames from TGVE-D (top) and TGVE-V (bottom), while the other columns present the corresponding frames edited by different methods. The complete videos are provided in the supplementary material.

frames edited by TokenFlow. In contrast, our method can generate consistent videos based on the prompt with flow-guided attention. More qualitative results are shown in Appendix B.

## 4.4 PLUG-AND-PLAY FLATTEN

FLATTEN can be seamlessly integrated into other diffusion-based T2V editing methods. To verify its compatibility, we incorporate FLATTEN into the U-Net blocks of ControlVideo (Zhang et al., 2023). The visual consistency of the videos edited by ControlVideo with FLATTEN is significantly improved, as shown in Figure 6. The fish (cyan box) in the bottom frame edited by the original ControlVideo disappears while using FLATTEN ensures a consistent visual appearance. We evaluate the ControlVideo with FLATTEN on TGVE-D. After integrating FLATTEN, the warping error $E_{warp}$ decreases remarkably from 6.81 to 4.78, while CLIP-T slightly decreases from 27.72 to 26.97. The editing score $S_{edit}$ is improved from **40.70** to **56.42**, which shows that FLATTEN can improve visual consistency for other T2V editing methods.

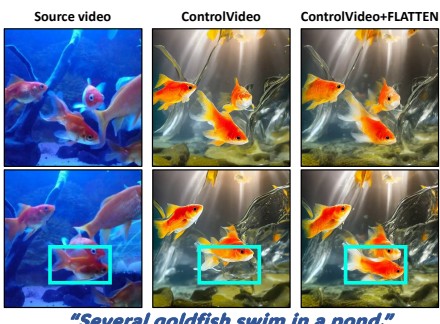

*"Several goldfish swim in a pond."*

**Figure 6:** FLATTEN can also improve visual consistency for other methods.

## 4.5 ABLATION STUDY

To verify the contributions of different modules to the overall performance, we systematically deactivated specific modules in our framework. Initially, we ablate both dense spatio-temporal attention (DSTA) and flow-guided attention (FLATTEN) from our framework. The dense spatio-temporal attention is replaced by the original spatial attention in the pre-trained image model. This is viewed as our baseline model (Base). As shown in Figure 7, the edited structure is sometimes distorted. We individually activate DSTA and FLATTEN. They both can reason about temporal dependencies and enhance structural preservation and visual consistency. As a further step, we combine

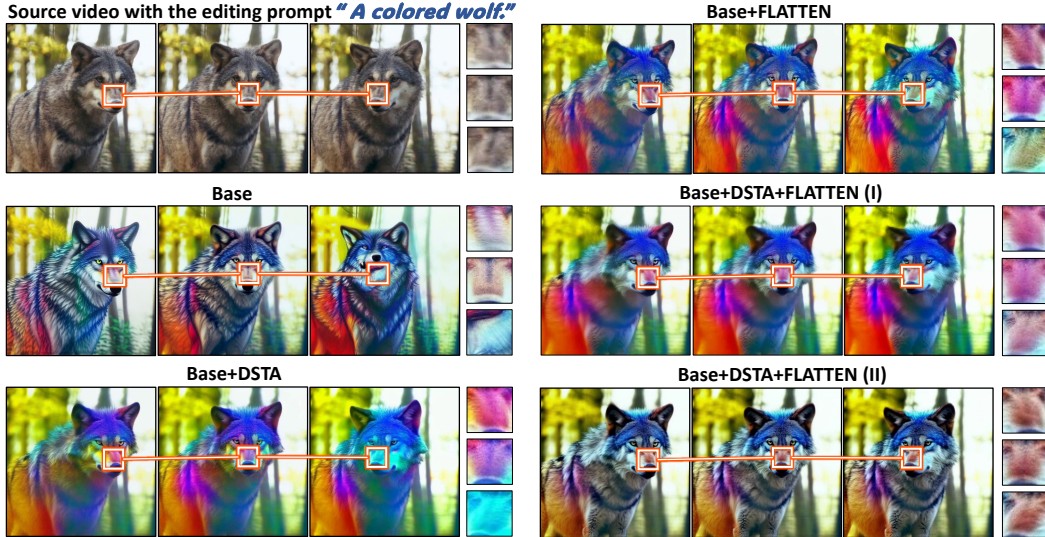

**Figure 7:** Qualitative results on the effectiveness of flow-guided attention (FLATTEN) and dense spatio-temporal attention (DSTA). We also explore two combinations of FLATTEN and DSTA. To easily compare visual consistency, we zoom in on the area of `nose` in different frames. In the lower right frames, both the structure as well as the colorization is temporally consistent.

**Table 2:** Ablation results for dense spatio-temporal attention (DSTA), flow-guided attention (FLATTEN), and their combinations on TGVE-D.

| Method | CLIP-T ↑ | Error$_{warp}$ ↓ | S$_{edit}$ ↑ |
|---|---|---|---|
| Base | 28.36 | 13.40 | 21.16 |
| Base + DSTA | 27.97 | 6.65 | 42.06 |
| Base + FLATTEN | 28.02 | 6.27 | 44.69 |
| Base + DSTA + FLATTEN (I) | 27.96 | 5.60 | 49.93 |
| Base + DSTA + FLATTEN (II) | 28.05 | 4.92 | 57.01 |

**Table 3:** User study of different T2V editing methods. The numbers indicate the average user preference rating (%).

| Method | Semantic | Consistency | Motion |
|---|---|---|---|
| Tune-A-Video | 18.43 | 7.42 | 8.18 |
| Text2Video-Zero | 11.01 | 4.49 | 4.21 |
| ControlVideo | 12.36 | 7.42 | 3.97 |
| FateZero | 8.09 | 13.26 | 17.76 |
| TokenFlow | 18.65 | 26.74 | 24.30 |
| FLATTEN (ours) | **31.46** | **41.12** | **41.59** |

DSTA and FLATTEN in two distinct ways and explore their effectiveness: (I) The output of dense spatio-temporal attention is forwarded to the linear projection layers to recompute the queries, keys, and values for FLATTEN; (II) The output of DSTA is directly used as queries, keys, and values for FLATTEN. We find that the first combination sometimes results in blurring, which reduces the editing quality. The second combination performs better and is adopted as the final solution. The quantitative results for the ablation study on TGVE-D are presented in Table 2.

## 4.6 USER STUDY

We conduct a user study since automatic metrics cannot fully represent human perception. We collect 180 edited videos and divide them into 30 groups. Each group consists of 6 videos edited by different methods with the same source video and prompt. We asked 16 participants to vote on their preference from the following perspectives: (1) **semantic** alignment (2) visual **consistency**, and (3) **motion** and structure preservation. The average user preference rating is shown in Table 3. Our method achieves higher user preference in all perspectives. More details are shown in Appendix C.

## 5 CONCLUSION

We propose FLATTEN, a novel flow-guided attention to improve the visual consistency for text-to-video editing, and present a training-free framework that achieves the new state-of-the-art performance on the existing T2V editing benchmarks. Furthermore, FLATTEN can also be seamlessly integrated into any other diffusion-based T2V editing methods to improve their visual consistency. We conduct comprehensive experiments to validate the effectiveness of our method and benchmark the task of text-to-video editing. Our approach demonstrates superior performance, especially in maintaining the visual consistency for edited videos.

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

## A  DDIM INVERSION WITH FLATTEN

Flow-guided attention (FLATTEN) can also improve the DDIM inversion process, which is critical in our T2V editing framework. We have validated the effectiveness of FLATTEN for the editing task in the ablation study (see Table 2). To further demonstrate that FLATTEN can contribute to high-quality latent noise estimation, we perform DDIM inversion on the source videos and reconstruct them using the U-Net with and without FLATTEN, respectively. When activating FLATTEN during DDIM inversion, more details in the source video can be restored, such as the eyes of the goldfish in Figure 8. Quantitatively, using FLATTEN results in higher scores for reconstruction metrics, with PSNR (peak signal-to-noise ratio) and SSIM (structural similarity index measure) reaching the values of 33.89dB and 0.9159, respectively. In contrast, PSNR and SSIM of the reconstruction without FLATTEN drop to 32.74dB and 0.8974. The quantitative results are shown in Table 4.

**Table 4:** The results of DDIM inversion and reconstruction with and without FLATTEN.

| Method | PSNR ↑ | SSIM ↑ |
|---|---|---|
| *w/o* FLATTEN | 32.74dB | 0.8974 |
| *w/* FLATTEN | 33.89dB | 0.9159 |

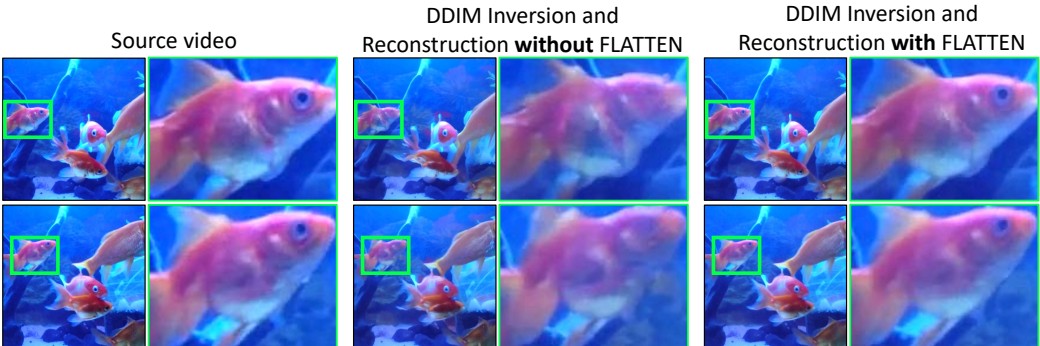

**Figure 8:** Using FLATTEN during DDIM inversion helps to improve the quality of the estimated latent noise. This is reflected in video reconstruction. The fish eyes and other details in the **third** column are successfully reconstructed, while in the **second** column, some details are missing.

## B  ADDITIONAL QUALITATIVE RESULTS

The additional qualitative results are shown in Figure 9 and Figure 11. With flow-guided attention, our training-free framework enables high-quality and highly consistent T2V editing.

To further demonstrate the visual consistency of videos generated by our approach, we provide the additional qualitative comparisons, which are shown in Figure 10. The videos produced by FLATTEN exhibit superior quality, characterized by a remarkable level of visual consistency and semantic alignment.

## C  USER STUDY DETAILS

We randomly sampled 30 source videos from TGVE-D and TGVE-V then edit them with 6 text-to-video editing approaches, including Tune-A-Video (Wu et al., 2022), FateZero (Qi et al., 2023), Text2Video-Zero (Khachatryan et al., 2023), ControlVideo (Zhang et al., 2023), ControlNet (Zhang & Agrawala, 2023), TokenFlow (Geyer et al., 2023) and our FLATTEN. For each group, we asked 16 participants to vote on their preference for 6 edited videos from the following perspectives:

- **Semantic** Alignment: The edited videos should match the given editing prompt.
- Visual **Consistency**: The adjacent frames in the edited videos should be smooth.
- **Motion** and Structure Preservation: The motion/structure of the edited videos should align with the source video.

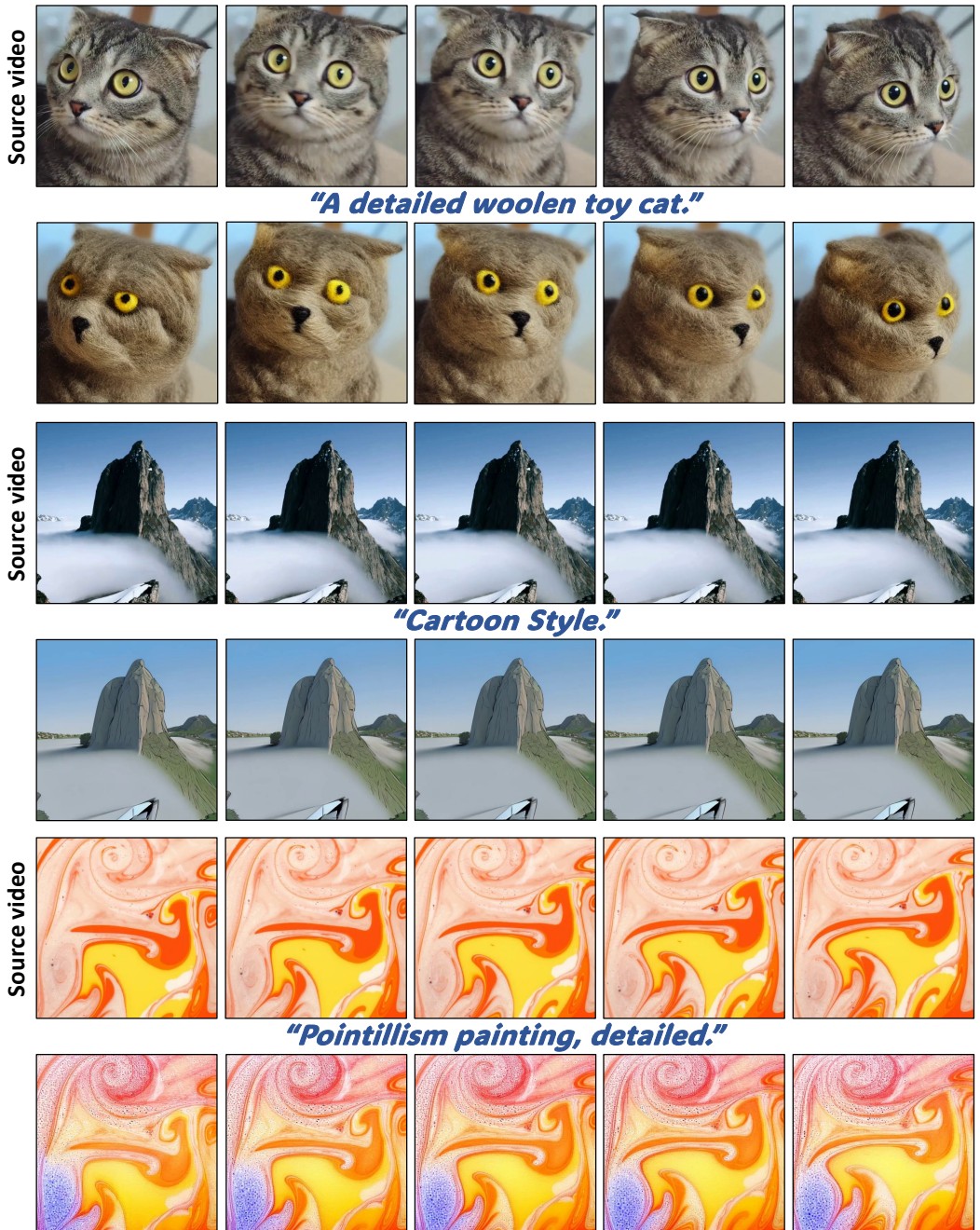

**Figure 9:** Additional qualitative results. The complete videos are provided in the supplementary material.

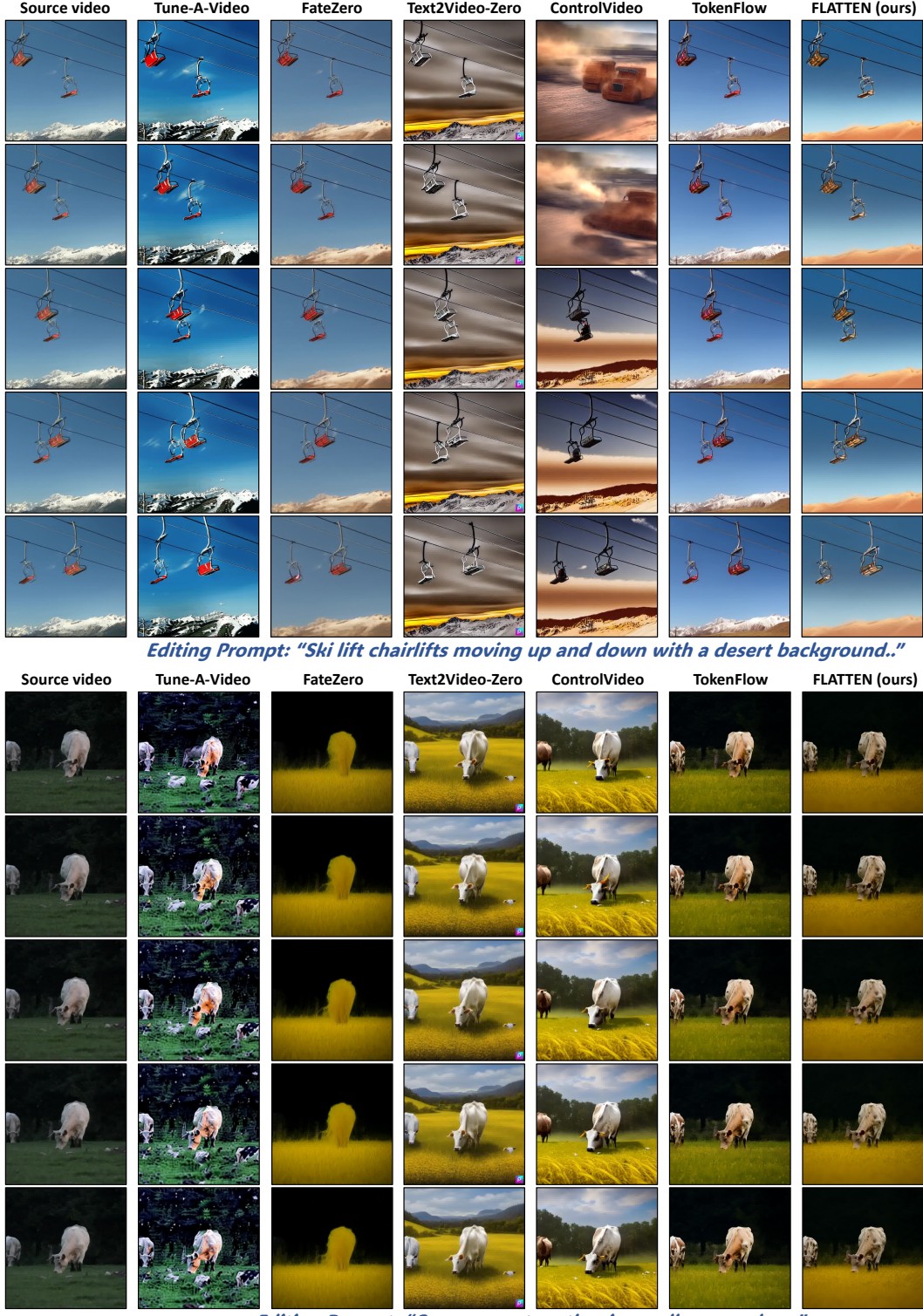

**Figure 10:** Qualitative comparison between advanced text-to-video editing approaches and FLAT-TEN.

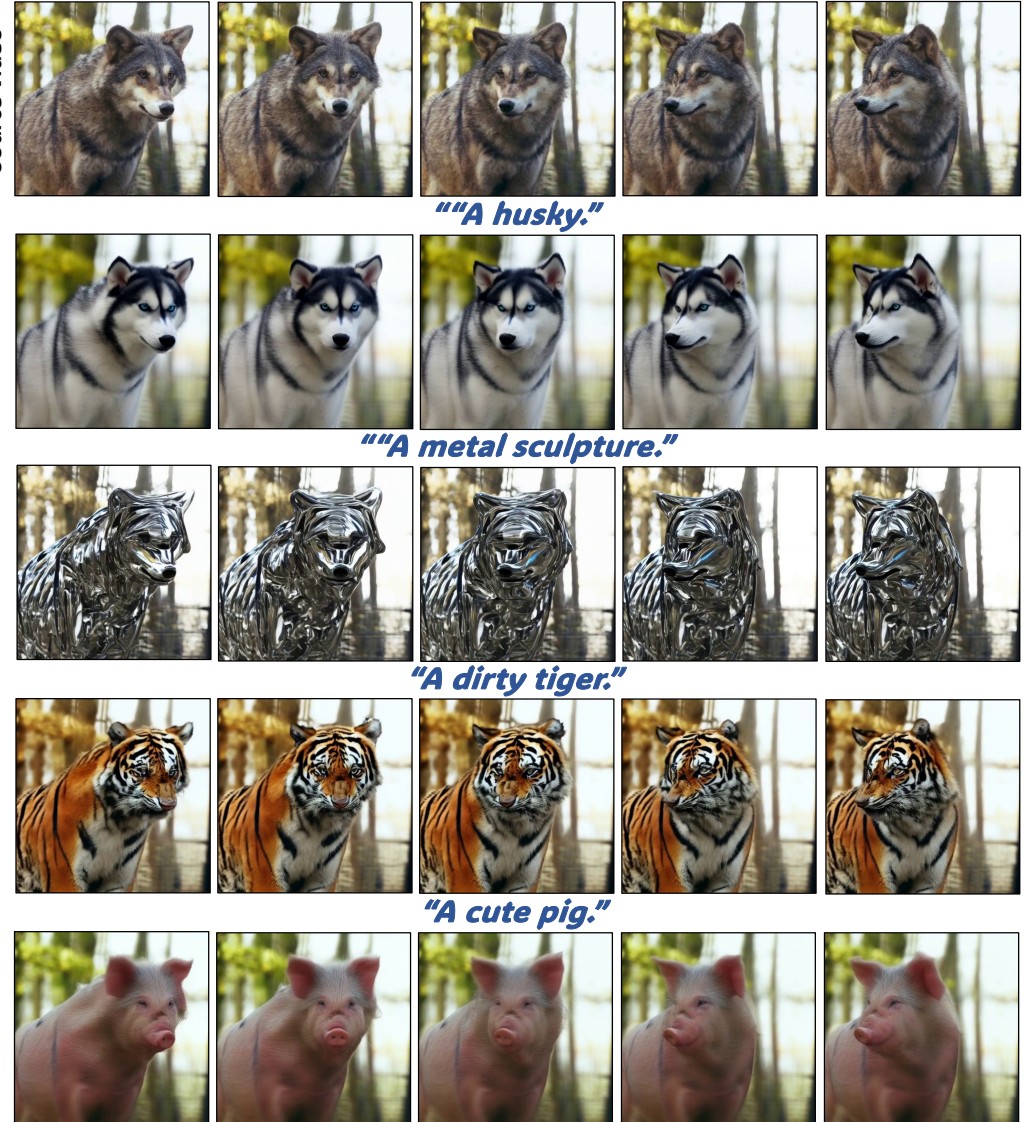

**Figure 11:** Our approach can output highly consistent videos conditional on different textual prompts.

An example of our user study interface is shown in Figure 12.

## D LIMITATIONS

Our approach is designed for highly consistent text-to-video editing utilizing optical flow from the source video. Therefore, our approach excels in style transfer, coloring, and texture editing but is relatively limited in dramatic structure editing. A failure case is demonstrated in Figure 13. The shape of sharks is completely different from quadrotor drones. The model changes the original sharks into "mechanical sharks", but not drones.

## E TRAJECTORY VISUALIZATION

The flow estimator, Raft (Teed & Deng, 2020), has demonstrated its superior performance in many applications, being able to accurately predict the flow field of dynamic videos. To demonstrate the

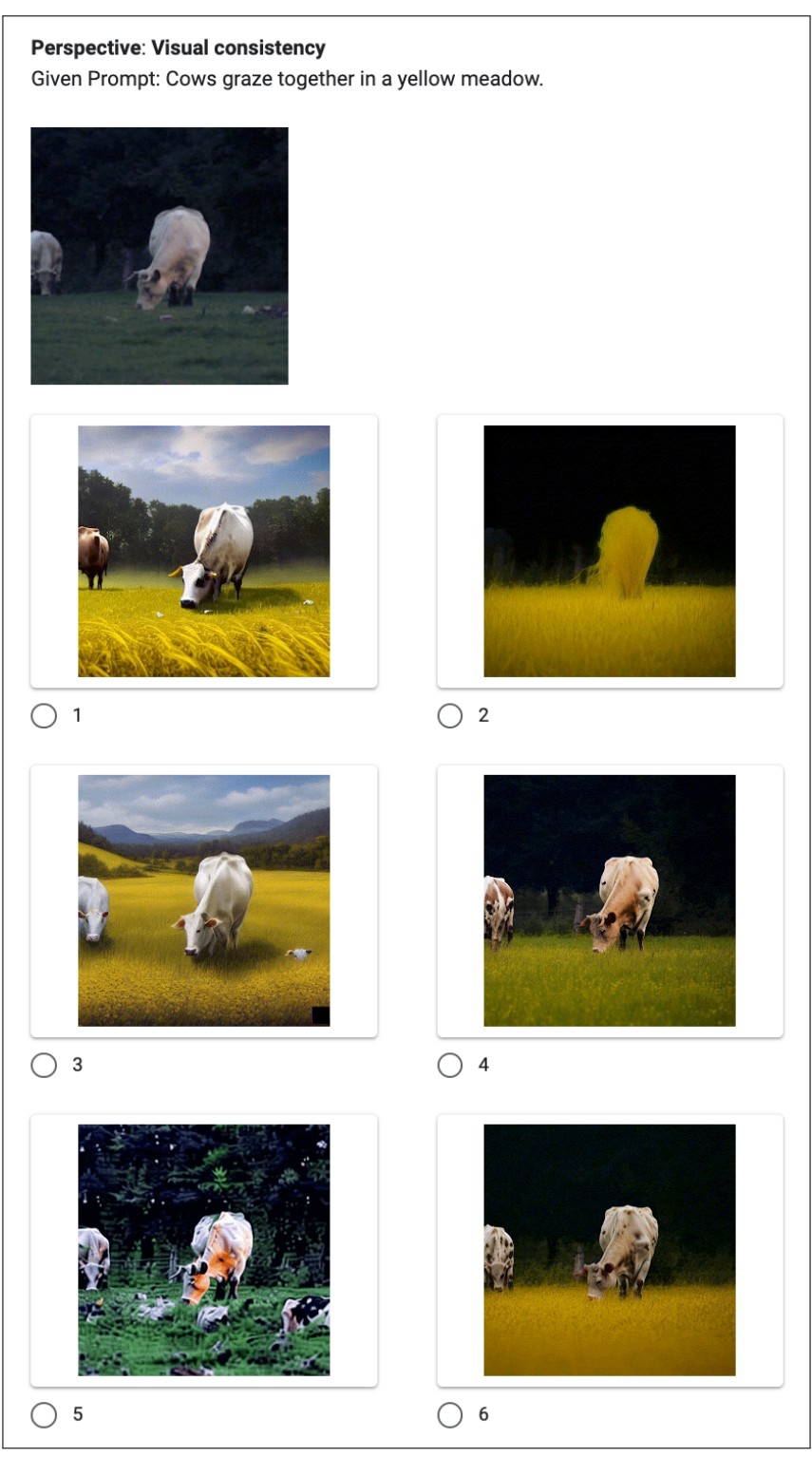

**Figure 12:** An example of our user study interface. Given a source video with an editing prompt, users should select their preferred video from 6 videos edited by different T2V editing methods from different perspectives (*e.g.,* visual consistency).

**Source video**                    **Edited video**

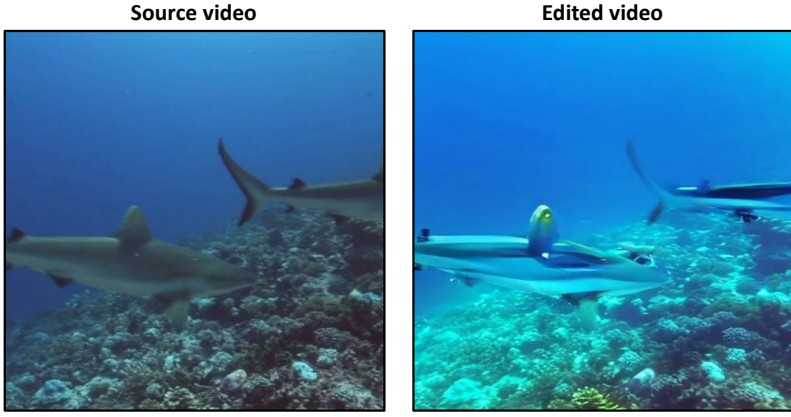

*" Two quadrotor drones swim in the blue ocean on a coral reef."*

**Figure 13:** Our approach is relatively limited in dramatic structure editing, *e.g.,* turning sharks into drones.

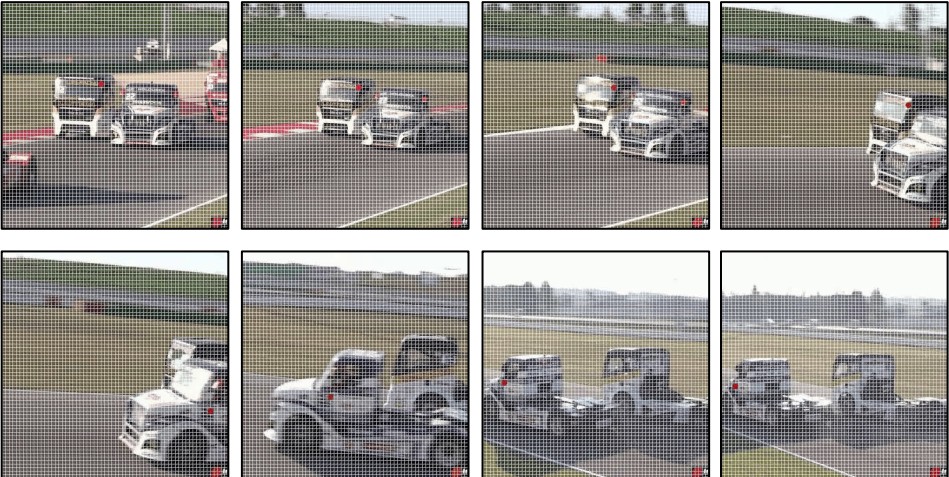

**Figure 14:** Visualization of the patch trajectories. The trajectories are computed based on the downsampled flow fields ($64 \times 64$) and the patches on the trajectories are marked with red dots.

robustness of the flow field estimation, we sample several predicted trajectories for video examples with large motion and visualize the trajectories in Figure 14. RAFT is robust even for videos with large and abrupt motions. Note that our approach does not rely on any specific flow estimation module. The trajectory prediction could be more precise with better flow estimation models in the future.

## F    ROBUSTNESS TO FLOWS

One notable advantage of our method is the integration of the flow field into the attention mechanism, significantly enhancing adaptability and robustness. To further demonstrate the robustness of FLATTEN to the pre-computed optical flows, we add random Gaussian noise to the pre-computed flow field and use the corrupted flow field for video editing. The qualitative comparison is shown in Figure 15. The corrupted flow field results in a few artifacts in the edited video (3rd row). However, the editing result is still better than the output of the baseline model without using optical flow as guidance.

Moreover, we replace the optical flow from RAFT in flow-guided attention with the flow estimated by another flow prediction model, GMA (Jiang et al., 2021). The comparison is shown in Figure 16. There is no obvious difference between the output videos and it shows that our method is robust to small differences in patch trajectories.

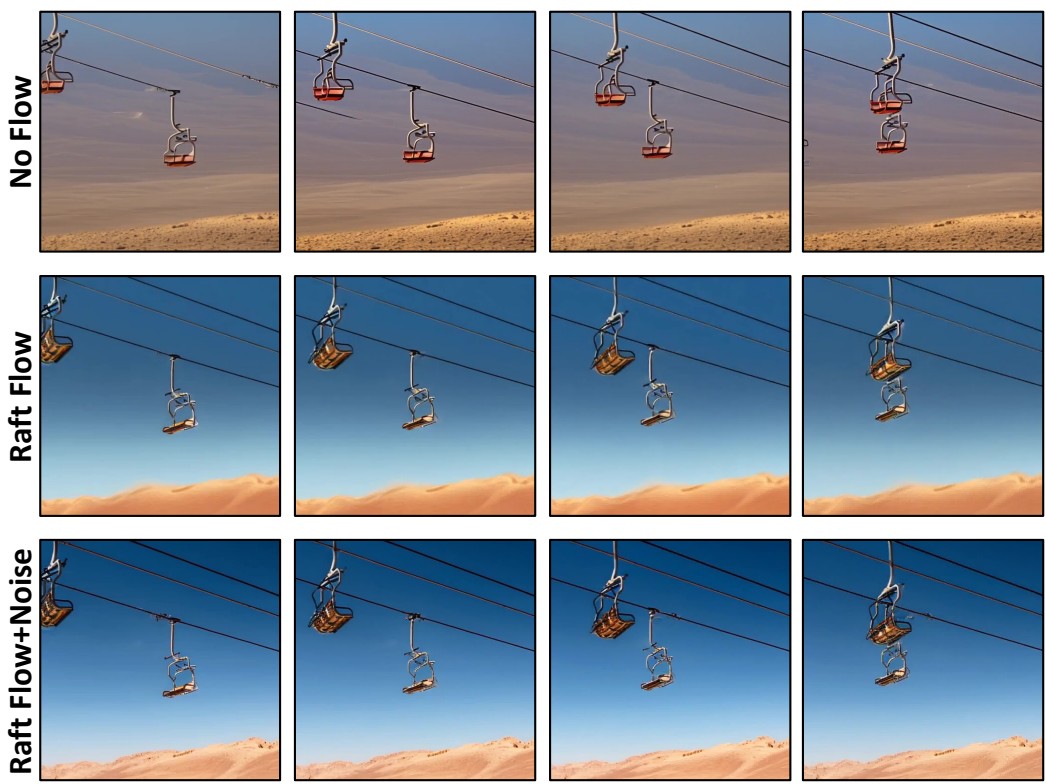

**Figure 15:** Video editing results from the baseline model (1st row), FLATTEN with the Raft flow (2nd row), and FLATTEN with the noised flow (3rd row).

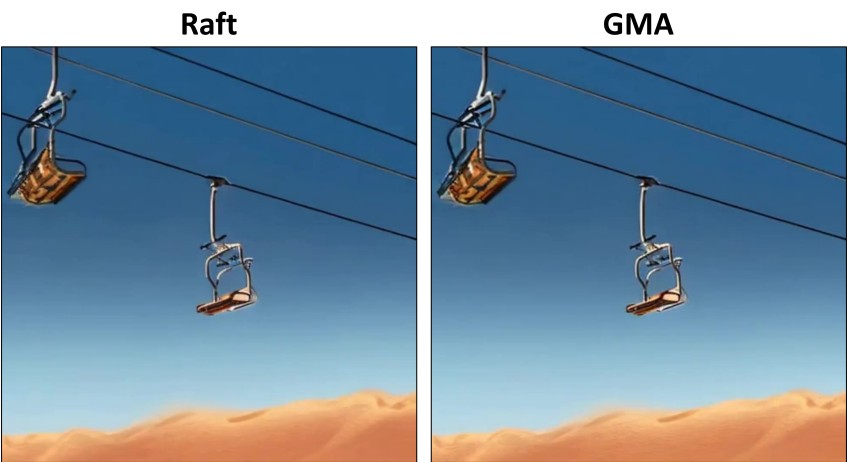

**Figure 16:** Comparison between using the optical flow from Raft (left) and GMA (right).

## G    RUNTIME EVALUATION

To compare the computational cost of different text-to-video editing models, we measure the runtime required to edit a single video (with 32 frames) by the different models. The runtime of the different models at different stages on a single A100 GPU is shown in Table 5. Our model has a relatively short runtime in the sampling stage and there is scope for further improvement.

**Table 5:** Runtime evaluation of different T2V editing models.

| Method | Finetuning | DDIM Inversion | Sampling |
|---|---|---|---|
| Tune-A-Video (Wu et al., 2022) | 11min15s | 3min52s | 3min34s |
| Text2Video-Zero (Khachatryan et al., 2023) | - | - | 3min17s |
| ControlVideo (Zhang et al., 2023) | - | - | 4min36s |
| FateZero (Qi et al., 2023) | - | 4min56s | 4min49s |
| TokenFlow (Geyer et al., 2023) | - | 3min41s | 3min29s |
| FLATTEN (ours) | - | 3min52s | 3min45s |

