# OpenReview forum: "FLATTEN: optical FLow-guided ATTENtion for consistent text-to-video editing"
_ICLR.cc/2024/Conference — ICLR 2024 poster_

### Official Review · Reviewer_Hpug · 2023-10-30

**Soundness:** 2 fair
**Presentation:** 3 good
**Contribution:** 3 good
**Rating:** 8
**Confidence:** 5

**Summary:**

The paper investigates the text-to-video editing task. To improve the temporal consistency, they introduce dense spatio-temporal attention and flow-guided attention to acquire information from the whole video. The proposed method achieves good performance on the test videos.

**Strengths:**

1. The proposed flow-guided attention is novel and effective.

2. The proposed flow-guided attention can be applied to other base models to further improve the temporal consistency.

3. Extensive experiments and ablation studies demonstrate the effectiveness of the proposed method.

**Weaknesses:**

1. The comparison with previous works is very limited. For the video editing task, multiple images should be shown for qualitative comparison in the main paper (Figure 5), which is important to verify the temporal consistency. Furthermore, even in the supplementary material, only one example is compared with previous works, which is not convincing. I compared the ``wolf’’ example with the TokenFlow example on its website. The results of this paper are not good to me.

2. From my perspective, the text information for editing is only acquired by cross attention with editing prompts. Other editing techniques like prompt-to-prompt are not used. This might cause the inaccurate editing of the background. For example, in the first example of Figure 5, the grass also turns yellow, while TokenFlow and FateZero can better keep the background.

3. DSTA conducts cross attention across all 32 frames, which takes many computational resources. It is better to compare the computational cost and inference time with other methods.

4. There are also some works based on optical flow trajectory for video generation [ref-1,ref-2,ref-3], which should also be discussed in the related work.

[ref-1] Dragnuwa: Fine-grained control in video generation by integrating text, image, and trajectory.

[ref-2] Generative image dynamics.

[ref-3] Motion-Conditioned Diffusion Model for Controllable Video Synthesis.

**Questions:**

The paper is good, but I still have some concerns as in the weaknesses.

---

> ### Author Response · Authors · 2023-11-15
> **Authors' response**
>
> **Dear Reviewer Hpug**,
>
> We appreciate your recognition of the novelty and effectiveness of our work! We will address your concerns as follows:
>
> **Q1: The comparison with previous works is very limited.**
>
> We release more qualitative comparisons at the anonymous link: https://youtu.be/dkXcjVrp9x0. We cannot show many examples in Figure 5 due to space limitations. Further examples will be added in Appendix B in the revised manuscript.
>
> Regarding the "wolf" example, we notice that the source video we used (19 frames/15FPS) is different from that used in the concurrent work TokenFlow (40 frames/20FPS). Our source video has fewer frames and lower FPS, which may impact the viewing experience. When compared frame by frame, our results are also of high quality (per-frame comparison:https://i.imgur.com/F7FHgWA.jpg).
>
> **Q2: The text information for editing is only acquired by cross attention. Other editing techniques like prompt-to-prompt are not used. This might cause the inaccurate editing of the background. E.g., the grass color changed in the truck editing example.**
>
> Our task focuses on general video editing rather than object-level editing. The influence of editing prompts on undescribed regions (e.g., the background in the “truck” example) is out of the scope of this work and should not be viewed as our weakness. We pursue semantic alignment and visual quality of the overall video instead of isolating edits to specific objects. We find background changes are beneficial as long as they contribute to the overall quality. For example, yellow grass serves as a more natural background than the original green grass for the editing target "wooden truck".
>
> We believe that other image editing techniques, such as prompt-to-prompt, can also be beneficial for text-to-video editing. However, in this paper, our objective is to address the inconsistency issue in text-to-video editing, which is more pressing for improving the quality of generated videos. On the other hand, since we did not make any changes to cross-attention, our framework is compatible with editing techniques such as prompt-to-promp. Btw, in the “truck” example, TokenFlow and FateZero failed to edit the trucks. Their output videos are quite similar to the source video and keep the background. These two works sometimes also change the background:
>
>  "cat ➜ red tiger" by FateZero: https://i.imgur.com/mwRX3vd.gif.
>
> The background is changed by TokenFlow: https://i.imgur.com/F7FHgWA.jpg
>
> **Q3: DSTA takes many computational resources. It is better to compare the inference time with other methods.**
>
> DSTA (dense spatio-temporal attention) is implemented using xFormers which can save a lot of computational resources. In fact, DSTA is also used in other works [1,2,3]. We have measured the inference time of different models for reference. Some models need finetuning and DDIM inversion before inference. The average time cost (for 32 frames) of different models is shown in the following table. Our model has a relatively short runtime in the sampling stage and there is scope for further improvement.
>
> |Model | Finetuning | DDIM Inversion | Sampling |
> |---|---|---|---|
> | Tune-A-Video  |11min15s | 3min52s  | 3min34s  |
> | Text2Video-Zero |  - | -  | 3min17s  |
> |  ControlVideo | -  |  - | 4min36s  |
> |  FateZero | -  |  4min56s  | 4min49s |
> |  TokenFlow | -  | 3min41s  | 3min29s |
> |  FLATTEN | -  |  3min52s  | 3min45s |
>
> [1] Controlvideo: Training-free controllable text-to-video generation.
>
> [2] Pix2Video: Video Editing using Image Diffusion
>
> [3] TokenFlow: Consistent Diffusion Features for Consistent Video Editing
>
> **Q4: There are also some works based on optical flow trajectory, which should also be discussed in the related work.**
>
> Thanks a lot for noticing these amazing works! We will include [ref-1,ref-2,ref-3] in Section 2 of the revised manuscript to strengthen our storyline.
>
> Your comments will help us to improve the presentation and impact of our work, and we are grateful for the opportunity to address these concerns. If you have any new comments, please do not hesitate to let us know! We appreciate your consideration of our contributions.

---

> > ### Comment · Reviewer_Hpug · 2023-11-22
> > **Response to the rebuttal**
> >
> > Dear authors,
> >
> > Thank you for the response and additional qualitative results.
> >
> > I hope to further clarify my concerns：
> >
> > - **The text information for editing is only acquired by cross attention.**
> > My concern is the textual alignment. The flow-guided attention has no influence on the textual alignment, which only relies on the cross-attention. Instead, techniques like prompt-to-prompt can emphasize the edited part. Therefore, the editing ability of purely cross-attention should be worse than such techniques.
> > However, in the results shown in the paper and the video, this paper seems to have better textual editing ability than TokenFlow (which uses prompt-to-prompt), such as the truck example. I have no idea the reason for it. I hope the authors can directly compare their method with the examples shown in the TokenFlow website under the same setting.
> >
> > - **Computational Cost.** Another main computational cost is the VRAM. TokenFlow applies DSTA only in the key frames while Pix2Video and ControlVideo only apply cross-frame attention to two reference frames, which is much fewer than the whole video (32 frames). Therefore, I think the VRAM requirement is still high even when using xFormers. Note that not everyone has 80GB A100 and keeping an acceptable VRAM requirement is also important.

---

> > > ### Author Response · Authors · 2023-11-22
> > > **Thanks for your reply!**
> > >
> > > **Dear Reviewer Hpug**,
> > >
> > > Thank you for your constructive feedback! We are glad to address your concerns within a limited time:
> > >
> > > **1. The text information for editing is only acquired by cross attention.**
> > >
> > > Now we fully understand your concern! We checked the TokenFlow paper carefully and we find that **TokenFlow didn’t use prompt-to-prompt** [1]. They use the image editing technique named PnP-Diffusion [2] (as their default setting: https://github.com/omerbt/TokenFlow/blob/master/configs/config_pnp.yaml). We follow their default setting when generating their samples. Maybe you're confused because of the acronym **PNP** and **prompt-to-prompt**? The editing ability of TokenFlow also purely relies on cross-attention. The better textual editing ability is probably because we don’t need to inject a lot of features/attention weights as TokenFlow to keep motion/structure for very dynamic source videos (e.g., the “truck”).
> > >
> > > Furthermore, we find that the design of visual attention can also affect textual cross-attention in the experiments. They are not completely independent. In the ablation study, we have tried two ways to integrate the flow-guided attention into the framework while keeping the other modules the same (see Figure 7 in the manuscript). However, even though the cross-attention modules are identical, the editing results are different.
> > >
> > > Following your suggestion, we use the same setting as their configuration (https://github.com/omerbt/TokenFlow/blob/master/configs/config_pnp.yaml) to edit their source video “woman-running.mp4”.
> > >
> > > Our result video: https://i.imgur.com/7ospOcz.mp4
> > >
> > > We cannot reproduce the result shown on their webpage (maybe they used a different configuration than they released). We followed their official code as well as their configuration (https://github.com/omerbt/TokenFlow/blob/master/configs/config_pnp.yaml) and got the result:
> > >
> > > Result video from TokenFlow: https://i.imgur.com/mapBHhK.mp4
> > >
> > > In this case, our result shows better structure preservation and a more stable background. We also would like you to consider the overall quantitative results. Moreover, since the concurrent work TokenFlow was released 2 months before ICLR submission (and the code was released 2 weeks before the submission), we ask for your understanding that a comprehensive comparison with it was not feasible at the point of submission.
> > >
> > > [1] Amir Hertz, et al. Prompt-to-prompt image editing with cross attention control.
> > >
> > > [2] Narek Tumanyan, et al. Plug-and-play diffusion features for text-driven image-to-image translation.
> > >
> > > **2. Computational Cost.**
> > >
> > > Thanks for your notice! The GPU that we used is A100 40GB which has 40 GB VRAM (usually used 27GB). We have carefully checked the code of ControlVideo. They also use DSTA for the whole video by default (Line447-448 https://github.com/YBYBZhang/ControlVideo/blob/master/models/attention.py). The reproduced results of ControlVideo in our paper also follow this default setting.  We also tried to run FLATTEN on RTX4090 which usually allows 27 frames. We will take your suggestion and improve the memory efficiency in the future to edit more frames on common GPUs! As DSTA is not our kernel idea, it can be easily replaced, e.g. by cross-frame attention to two/four/... adjacent frames. We sincerely hope that you will consider our main contribution, which is effective and efficient. **Our flow-guided attention (FLATTEN) has great potential in this case.** You see, we limit spatial attention in FLATTEN while allowing attention to look farther in time, thus saving memory and computation.
> > >
> > > Again, we sincerely thank you for your valuable time and constructive comments! We appreciate your consideration of our contributions!
> > >
> > > Best regards,
> > >
> > > Paper 1423 Authors

---

### Official Review · Reviewer_F5Lm · 2023-10-30

**Soundness:** 2 fair
**Presentation:** 3 good
**Contribution:** 2 fair
**Rating:** 5
**Confidence:** 2

**Summary:**

This paper proposes text-guided video editing systems that considers optical flow to preserve temporal consitency
In detail the temporal attention is guided by the paths estimated from optical flow.

**Strengths:**

[+] The idea makes sense that involves optical flow into the diffusion model for holding temporal consistency
[+] Performances are enhanced compared to previous editing systems

**Weaknesses:**

[-] Optical flow can be effective when a single objective appears. However, we can easily come up with other cases including occlusion, objects' appearing or disappearing, or else. Therefore the method seems sensitive to the input video, which can ruin the attention or even worse than previous temporal attention methods. Can you explain why the method should be better than previous temporal attention?

[-] The performances are better than many previous works. What are the samples that this method validates? I am quite curious about the videos that they evaluated.

[-] Are there any qualitative or quantitative results about the trajectory patches in the aforementioned cases in question [1]? I want to see if the trajectory is truely following the flow of objectives in the video.

**Questions:**

My questions are above

---

> ### Author Response · Authors · 2023-11-15
> **Authors' response**
>
> **Dear Reviewer F5Lm**,
>
> We appreciate your recognition of our work in enhancing temporal consistency! We will address your concerns as follows:
>
> **Q1: We can come up with other cases including occlusion, object appearing or disappearing, or else. Therefore the method seems sensitive to the input video. Can you explain why the method should be better than previous temporal attention?**
>
> Your concern about sensitivity to input video is valid, and we acknowledge the complexity introduced by occlusion, object appearing or disappearing. In fact, our flow-guided attention is proposed to address these issues, where the previous temporal attention may be problematic.
>
> In contrast to previous temporal attention techniques that lack explicit motion guidance, our approach utilizes optical flow information to guide attention. Note that when the occlusion, object appearing/disappearing, the corresponding patch trajectories are also created (e.g., appearing) or stopped (e.g., disappearing). In the previous temporal attention, the query patch attends to all other patches in the video and aggregates their features. The irrelevant patches can mislead the attention process.
>
> For example, when an object disappears in the (t+1) frame, the object patch in the (t) frame still attends to the patches in the (t+1) frame. For our flow-guided attention, the trajectory of the object patch stops at the (t) frame and the object patch in the (t) frame doesn’t attend to the patches in the (t+1) frame. The robustness of our flow-guided attention is derived from its ability to interpret complex motion patterns, addressing the limitations of previous temporal attention.
>
> **Q2: The performances are better than many previous works. What are the samples that this method validates? I am quite curious about the videos that they evaluated.**
>
> For the quantitative study, we use the same evaluation set (LOVEU) as described in the main paper. The videos and prompts used for the quantitative results can be found at: https://sites.google.com/view/loveucvpr23/track4.  We also use some in-the-wild videos for qualitative results (e.g. the cute cat in Figure 1).
>
> **Q3: Are there any qualitative or quantitative results about the trajectory patches?**
>
> Sure! The patch trajectories are predicted well even in complex scenarios. The qualitative results are shown as follows (Reviewer UKpY is also interested in).  The patches on the trajectories are marked with red dots. Note that our approach does not rely on any specific flow estimation module. We use the SOTA flow estimation module RAFT [1] in this work, and the trajectory could be more precise/robust with better flow models in the future.
>
> Occlusion: https://i.imgur.com/8PErmVI.gif
>
> Appearing: https://i.imgur.com/R22S7Ri.gif
>
> Disappearing: https://i.imgur.com/w3nRRwK.gif
>
> Large motion: https://i.imgur.com/fmyPK9q.gif
>
> [1] Raft: Recurrent all-pairs field transforms for optical flow.
>
> Your comments will help us to improve the presentation and impact of our work, and we are grateful for the opportunity to address these concerns. If you have any new comments, please do not hesitate to let us know! We appreciate your consideration of our contributions.

---

### Official Review · Reviewer_1FeZ · 2023-10-31

**Soundness:** 2 fair
**Presentation:** 3 good
**Contribution:** 2 fair
**Rating:** 5
**Confidence:** 4

**Summary:**

To improve the visual consistency for text-to-video editing， FLATTEN is proposed to enforce the patches on the same flow path across different frames to attend to each other in the attention module. Experiment results on existing text-to-video editing benchmarks show that the proposed method achieves the new state-of-the-art performance.

**Strengths:**

The proposed Flow-guided attention is intuitive and makes sense.

**Weaknesses:**

1. The method is only suitable for scenarios where every pixel in the original video aligns spatially with the generated video. For misaligned areas, the optical flow trajectory of the original video is not appliable for the motion of the generated video, leading to incorrect key and query identifications. For instance, in the example of transforming a cat to a tiger in Figure 1, the tiger's face is larger than the cat and thus there exist pixels that belong to the tiger's face and belong to the background in the cat example. For the original video with the cat, these pixels belong to the background with an optical flow near zero. However, for the tiger, it's part of the face and should rotate with the head, requiring an optical flow describing a leftward movement. This sets too high a requirement for editing scenarios.

2. If optical flow tracking is accurate enough, why not simply select a keyframe and then directly copy pixels following the same optical flow path to other frames? This approach seems more accurate than aligning through attention. For example, bilinear interpolation combined with optical flow is often used to predict the next frame in videos.

3. In the provided MP4, there is only one visual comparison, which is too limited. It's suggested not to cherry-pick comparisons so that the effectiveness of the method can be judged intuitively.

**Questions:**

See Weakness.

---

> ### Author Response · Authors · 2023-11-15
> **Authors' response**
>
> **Dear Reviewer 1FeZ**,
>
> We appreciate your insightful thoughts on the application scenarios of our work! We will address your concerns as follows:
>
> **Q1: The method is only suitable for scenarios where every pixel in the original video aligns spatially with the generated video. E.g., in the example of transforming a cat to a tiger in Figure 1.**
>
> Our method intentionally avoids the requirement for pixel-level alignment between the original and generated videos. The optical flow is downsampled before being used as guidance (e.g. from 512 * 512 to 64 * 64) and the flow-guided attention works in low-resolution latent space. This improves the tolerance for misaligned areas between the original video and the generated video. In the example of “cat to tiger”, even though the tiger's face is larger than the original cat's. The pixels that belong to the tiger's face move naturally in the generated video and the tiger's face does not stick to the background.
>
> The quantitative and qualitative results demonstrate that our model can output high-quality and highly consistent videos. The edited videos by FLATTEN have natural visual appearance and the same dynamic motion as the original videos. This allows our method to be credibly applied in many application scenarios including style transfer, coloring, texture editing, and shape editing (e.g., editing short videos on social media platforms).
>
> Your concern about misaligned areas is insightful! In fact, it is challenging not only for our method but also for all current video editing approaches. The other methods using optical flow [1,2] or structural information (e.g., edge/depth map) [3,4] from the original video also face this misalignment problem. Tune-A-Video [5] with finetuning has tried the large-shape editing but the original motion is lost in the result (e.g., https://i.imgur.com/NVT5QbY.gifv). For the task of video editing, the edited video should follow the motion/structure from the source video. Large misaligned areas are somehow in conflict with this task. This issue might be better solved using video generation techniques.
>
> [1] VideoControlNet: A Motion-Guided Video-to-Video Translation Framework by Using Diffusion Model with ControlNet
>
> [2] Tokenflow: Consistent diffusion features for consistent video editing.
>
> [3] Text2video-zero: Text-to-image diffusion models are zero-shot video generators.
>
> [4] Controlvideo: Training-free controllable text-to-video generation.
>
> [5] Tune-a-video: One-shot tuning of image diffusion models for text-to-video generation.
>
> **Q2: If optical flow tracking is accurate enough, why not simply select a keyframe and then directly copy pixels (e.g. bilinear interpolation) following the same optical flow path to other frames?**
>
> While the idea of directly copying patches based on optical flow seems intuitive, our flow-guided attention in a latent space provides a more adaptive solution. First, selecting keyframes introduces additional complexity. The robustness of the model might be influenced by the effectiveness of the keyframe selection strategy. Moreover, even if a suited keyframe is successfully chosen, the patch embeddings are not easy to propagate in the latent space. In fact, we found that using bilinear interpolation for patch prediction often leads to high-frequency flickering, which introduces undesired artifacts, compromising the visual quality of the generated frames (see the highlighted region in the example at: https://i.imgur.com/OUIeHr9.png). In contrast, our flow-guided attention dynamically adjusts to complex scenes and provides a more robust and adaptive approach to frame alignment. We will stress this aspect in the revised manuscript.
>
> **Q3: In the provided MP4, there is only one visual comparison, which is too limited.**
>
> The supplementary video is only a brief introduction and there is an upload limitation (<100MB). Therefore we only showed a challenging example with larger motion. We randomly sample more examples and release them at the anonymous link: https://youtu.be/dkXcjVrp9x0. Further examples will be added to Appendix B in the revised manuscript.
>
> Your comments will help us to improve the presentation and impact of our work, and we are grateful for the opportunity to address these concerns. If you have any new comments, please do not hesitate to let us know! We appreciate your consideration of our contributions.

---

### Official Review · Reviewer_UKpY · 2023-10-31

**Soundness:** 4 excellent
**Presentation:** 3 good
**Contribution:** 2 fair
**Rating:** 8
**Confidence:** 4

**Summary:**

Summary: The paper focuses on text guided video editing. Previous methods to tackle this problem extend the text-to-image U-net to the temporal dimension to implement spatiotemporal attention where patches from different frames attend to one another. The paper argues that such methods introduce irrelevant information, since they allow all patches in the video to attend to one another where in fact many of these spatiotemporal patch to spatiotemporal patch connections might be irrelevant. To address this problem, the paper suggests using optical flow to guide the attention. Specifically, a pre-trained optical flow network is used to estimate the flow field and tracks of patches along flow trajectories are aggregated to enforce only patches on the same trajectory to attend to one another in a second step of MHSA. This results in more visual-consistent videos as the paper demonstrate both qualitatively and qualitatively

Method: First, the "standard" the text to image U-net architecture is inflated to account for the temporal dimension and the image patch spatial self attention mechanism is replaced with spatiotemporal self-attention with all patches in the video used as tokens for Q,K,V. Secondly, a pre-trained optical flow network is employed to compute the flow field along the frames of the video. Tracks of patches (in the latent space) are aggregated using the downsampled flow field. Next, self attention is performed between patch-embeddings on the same track. Specifically, the queries are taken from the original dense spatiotemporal MHSA, but for every query associated with a specific patch - the keys and values in MHSA are only the ones which are associated with patches on the same track. Note that this method does not require re-training as it only refines the existing embedded spatiotemporal patch embedded tokens with additional information by applying MHSA again but with restrictions on which patches can attend to one another (where this "restriction" is derived from the flow field).

Experiments: the paper compares the proposed method against 5 publicly available text-to-video editing methods on two standard benchmarks. The proposed method performs favourably both in terms of visual quality/alignment metrics  and visual consistency metrics. The paper also presents quantitative results as well as a user-study demonstrating the effectiveness of the method, particularly in the aspect of visual-consistency when motion is introduced.

**Strengths:**

The proposed method is sound and original. The framework is very simple, does not require further training and can be easily plugged to various existing architectures. The paper is well written and the effectiveness of the method is demonstrated relatively well.

**Weaknesses:**

In my opinion, a drawback of the method is the heavy reliance on pre-computed flow field using a pre-trained network that is used as a black box. Thus, errors in this step can negatively affect the results of the proposed pipeline. However, the paper does not address this issue and there are no results to measure the robustness of the method. See question in the section below.

**Questions:**

As I understand, the method is designed to improve visual consistency, particularly with respect to motion. The method relies on optical flow to "enhance" the embedded tokens with motion information derived from the flow field. As the pipeline relies on flow field computation and errors introduce in that step may affect the results. Something that is missing in the paper in my opinion is some discussion/experiments/results on how robust is the method to mistakes in the flow field computation. Specifically:

1. How well the proposed method can handle large motion (large displacement in the flow field) or abrupt motion? are there any examples you can provide?
2. How well the proposed method can handle videos in which both global motion (camera movement) and local motion (object movement) are present? are there any examples you can provide?
3. Are there any situations where the method can do "more harm than good"? I mean, cases where the errors in the flow-field computation can cause the method to produce worse results than the baseline? how often do they occur?
4. Are there any examples that you can provide in which the flow field is far from accurate? In those cases, are the results worse than the baseline, meaning the method did "more harm than good"?
5. How does the results change with respect to accuracy of the flow field? For example, by taking a specific video and flow field results from several models where some perform dramatically worse than others or by gradually corrupting the flow field and measuring the affect on the results?

I find that the qualitative results provided the supplemental video are extremely helpful (particularly the "racing trucks" example in which the results of other methods are provided). I would be grateful if the authors would be able to provide more examples that address the questions above.

---

> ### Author Response · Authors · 2023-11-15
> **Authors' response (1/2)**
>
> **Dear Reviewer UKpY**,
>
> We appreciate your positive recognition of our work and the insightful comments regarding the reliance on the pre-computed flow field. We will address your comments as follows:
>
> We would like to clarify that our method does not directly employ the flow field on the pixel level. Instead, we utilize a downsampled flow field as guidance in the latent space and incorporate an attention mechanism. This can effectively mitigate the negative impact of potential errors in the pre-computed pixel displacement field, which is also one key novelty of our work. Furthermore, the flow estimator [1] we used has already demonstrated its superior performance in many applications. To strengthen our idea, we will add the results/answers to the following questions in the revised manuscript.
>
> [1] Raft: Recurrent all-pairs field transforms for optical flow.
>
> **Q1: How well the proposed method can handle large motion or abrupt motion?**
> The proposed method based on RAFT overall performs well, even when there is a large motion/abrupt motion. In fact, "racing trucks" is the most challenging example in the dataset. It is a long-term video with large motion, camera movement, and object movement. Our model can still output a high-consistent edited video. The generated video has the same motion as the input video. Several patch trajectories in the example are visualized (the patches on the trajectory are marked with red dots) as follows:
>
>  https://i.imgur.com/8PErmVI.gif
>
>  https://i.imgur.com/R22S7Ri.gif (starts in the middle)
>
> Sometimes some patches (e.g. the background) look quite similar and the predicted trajectories are not 100% accurate. However, they have little negative impact on the editing results and our method is robust in these cases.
>
> https://i.imgur.com/bpHWf7k.gif
>
> We also show some additional editing results for the large motion videos:
>
> https://i.imgur.com/zlGGXpP.mp4
>
> https://i.imgur.com/D53sbDA.mp4
>
> https://i.imgur.com/fQQ8AOP.mp4
>
> **Q2: How well the method can handle videos in which both camera and object movement are present?**
>
> Our method works well with videos where camera movement and object movement occur at the same time. "Racing trucks" is a good example. Another example of “car drift” is shown:
>
> Source video: https://i.imgur.com/qN3bsGS.mp4
>
> Edited video: https://i.imgur.com/Sc81qzN.mp4
>
> Trajectories:
>
>  https://i.imgur.com/fmyPK9q.gif
>
> https://i.imgur.com/w3nRRwK.gif  (stops in the middle)
>
> https://i.imgur.com/0Y37Exi.gif  (starts in the middle)
>
> **Q3: Are there any situations where the method can do "more harm than good"? how often do they occur?**
>
> Our method demonstrates a high level of robustness, and instances where it causes "more harm than good" are very rare. While it's acknowledged that occasional inaccuracies may arise in the flow-field computation, it's crucial to emphasize that the majority of trajectories are accurately predicted. Importantly, the occurrence of all trajectories in a video being inaccurately predicted is an extremely rare anomaly that has not been observed in our extensive testing.
>
> One notable advantage of our approach is the integration of the flow field into the attention mechanism, significantly enhancing adaptability and robustness. In situations where a patch is mistakenly assigned to a trajectory, the attention mechanism mitigates potential harm. When a patch is misassigned to a trajectory, other patches will avoid attending to the wrong patch due to the low similarity with the misassigned patch. Moreover, if the original motion is very complex and many predicted trajectories are not correct, it usually means the original video is also very challenging for the baseline method (which only uses spatial-temporal attention). The results in Table 2 have demonstrated the robustness/effectiveness of our method compared to the baseline.

---

> ### Author Response · Authors · 2023-11-15
> **Authors' response (2/2)**
>
> **Q4: Are there any examples in which the flow field is far from accurate? In those cases, are the results worse than the baseline?**
>
> Since we don’t have the ground truth field, it is difficult to determine whether the pre-computed flow field is far from accurate. As discussed, from the qualitative results, most of the precomputed flow fields are of good quality. Therefore, we follow your suggestion in Q5: we corrupt the pre-computed flow field (by adding noise) and use the corrupted flow field to predict the patch trajectories for flow-guided attention. The results are available at:
>
> Source video (prompt: Ski lift chairlifts with a desolate desert background): https://i.imgur.com/ES7ab8T.mp4
>
>
> Baseline video: https://i.imgur.com/IhBWAKI.mp4
>
> RAFT flow: https://i.imgur.com/XeaTSwV.mp4
>
> RAFT flow+noise: https://i.imgur.com/BdCaOTI.mp4
>
> After adding the noise to the pre-computed flow, the trajectories are not accurate. There are artifacts in the generated video. However, the video quality is still better than the baseline model.  Moreover, we emphasize that our approach does not rely on any specific flow estimation model, e.g., RAFT. The robustness of flow estimation is out of the scope of this work.
>
>
> **Q5: How do the results change with respect to the accuracy of the flow field?**
>
> We use the optical flow computed by RAFT and GMA in our flow-guided attention. There is no obvious difference between the output videos:
>
> RAFT:  https://i.imgur.com/XeaTSwV.mp4
>
> GMA: https://i.imgur.com/qrdUFM4.mp4
>
> The results show that our method is robust to small differences in patch trajectories..
>
> Your comments will help us to improve the presentation and impact of our work, and we are grateful for the opportunity to address these concerns. If you have any new comments, please do not hesitate to let us know! We appreciate your consideration of our contributions.

---

### Author Response · Authors · 2023-11-18
**General Response**

**Dear Reviewers and ACs**,

Thank you so much for your time and efforts in assessing our paper! The insightful comments are very helpful in improving our work, and we have responded to the concerns point by point.

According to our understanding, there are two common concerns from reviewers:
- **How robust is FLATTEN to the flows?** One notable contribution of our approach is the integration of the optical flows into the attention mechanism, significantly enhancing adaptability and robustness. We utilize downsampled flow fields as guidance in the latent space to mitigate the negative impact of potential errors in the pre-computed flows and avoid the strict requirement for pixel-level alignment. A detailed analysis is provided to demonstrate the effectiveness of our method.
- **Is there more qualitative comparisons?** We provide more qualitative comparisons with previous works, available at the anonymous link: https://youtu.be/dkXcjVrp9x0.

We have taken all the suggestions carefully and updated our previous version. In the revised manuscript, we have made the following **changes** and highlighted them in blue:

1. Added the related works about controllable video generation using optical flows in Section 2.
2. Added more qualitative comparisons with other T2V editing methods in Appendix B.
3. Added the patch trajectory visualization in Appendix E.
4. Added the discussion about the robustness of FLATTEN to computed optical flows in Appendix F.
5. Added the runtime evaluation in Appendix G.

We are happy to discuss with you further if you still have other concerns. Thank you very much again!

Best regards,

Paper 1423 Authors

---

### Author Response · Authors · 2023-11-20
**Sincerely looking forward to your feedback**

**Dear Reviewers,**

We sincerely appreciate your great efforts in reviewing this paper. Your constructive advice and valuable comments really helped improve our work. Considering the approaching deadline, please, let us know if you have follow-up concerns. We hope you can consider our responses in your assessment, and we can further address unclear explanations and remaining concerns if any.

Once more, we appreciate the time and effort you've dedicated to our paper!

Best regards,

Paper 1423 Authors

---

### Meta-Review · Area_Chair_fnYH · 2023-12-06

**Metareview:**

This paper presents a novel method that utilizes optical flow to guide the attention in text-to-video editing.  With the assistance of a pre-trained network to calculate the optical flow field, the FLATTEN could be extended to multiple scenarios without further training. Extensive experiments and examples testify to its performance.  In the discussion phase, this work received mixed ratings. The authors replied to all concerns about limited comparisons and the robustness of this method, providing more examples with anonymous links. Considering the difficulty of the text-to-video task,  as well as the confidence of all reviews, I lean towards acceptance.

**Justification For Why Not Higher Score:**

This method has a high reliance on the pre-trained network to generate the optical flow field.

**Justification For Why Not Lower Score:**

Experiments and examples demonstrate its performance.

---

### Decision · Program_Chairs · 2024-01-16

Accept (poster)